# Effect of Devised Simultaneous Physical Function Improvement Training and Posture Learning Exercises on Posture

**DOI:** 10.3390/healthcare11091287

**Published:** 2023-04-30

**Authors:** Naonobu Takahira, Sho Kudo, Mako Ofusa, Kenta Sakai, Kouji Tsuda, Kiyoshi Tozaki, Yoshiki Takahashi, Hiroaki Kaneda

**Affiliations:** 1Sensory and Motor Control, Graduate School of Medical Sciences, Kitasato University, 1-15-1 Kitasato, Minami-ku, Sagamihara-shi 252-0373, Japan; gjiang214@gmail.com (S.K.); boku.ketaemon0111@gmail.com (K.S.); kouji.tsuda@ompu.ac.jp (K.T.); kiyoc1230@gmail.com (K.T.); t.yoshiki321@gmail.com (Y.T.); hiroaki.kh0821@gmail.com (H.K.); 2Physical Therapy Course, Department of Rehabilitation, School of Allied Health Sciences, Kitasato University, 1-15-1 Kitasato, Minami-ku, Sagamihara-shi 252-0373, Japan; zhenzidafang3@gmail.com; 3Department of Rehabilitation, Zama General Hospital, 1-50-1 Soubudai, Zama-shi 252-0011, Japan; 4Department of Rehabilitation Medicine, St. Marianna University School of Medicine Hospital, 2-16-1 Sugao, Miyamae-ku, Kawasaki-shi 216-8511, Japan; 5Department of Hygiene and Public Health, Osaka Medical and Pharmaceutical University, 2-7 Daiga-kumachi, Takatsuki-shi, Osaka 596-8686, Japan; 6Department of Rehabilitation, Juntendo University Urayasu Hospital, 2-1-1 Tomioka, Urayasu-shi 279-0021, Japan

**Keywords:** posture, crossed syndrome, physical training, posture learning, sustained effect

## Abstract

Poor posture in young adults and middle-aged people is associated with neck and back pain which are among the leading causes of disability worldwide. Training posture maintenance muscles and learning about ideal posture are important for improving poor posture. However, the effect of using both approaches simultaneously has not been verified, and it is unclear how long the effects persist after the intervention. Forty female university students were randomly and evenly assigned to four groups: physical function improvement training, posture learning, combination, and control groups. Four weeks of intervention training was conducted. Postural alignment parameters were obtained, including trunk anteroposterior inclination, pelvic anteroposterior inclination, and vertebral kyphosis angle. Physical function improvement training for improving crossed syndrome included two types of exercises: “wall-side squatting” and “wall-side stretching”. The posture learning intervention consisted of two types of interventions: “standing upright with their back against the wall” and “rolled towel”. A multiple comparison test was performed after analysis of covariance to evaluate the effect of each group’s postural change intervention on postural alignment. Only the combination group showed an effective improvement in all posture alignments. However, it was found that a week after the 4-week intervention, the subjects’ postures returned to their original state.

## 1. Introduction

Poor posture in young adults and middle-aged people is associated with neck and back pain which are among the leading causes of disability worldwide. In recent years, it has been reported that Cross syndrome can be caused not only in the work environment, but also in young and middle-aged people due to the persistence of specific repetitive postures in the student learning environment [1,2,3]. It is important to treat these problems from a preventive perspective, since poor posture from a young age can result in lost work opportunities.

Locomotive syndrome is the condition of being at risk of needing long-term care due to musculoskeletal disorders; it is reported among the aging population in Japan [4]. Those with locomotive syndrome often have weaker back muscles than those without the condition. Decreased back muscle strength leads to increased spine tilt and kyphosis [5], which are related to falls among the elderly [6]. Poor posture can also lead to poor gait, functional performance, and balance [7]. It is important to correct posture from a young age from a preventive perspective because poor posture can cause falls.

In the ideal standing position, the center of gravity passes vertically through the earlobe, acromion, greater trochanter, anterior knee joint (posterior to the patella), and anterior part of the lateral malleolus. In the frontal plane, the center of gravity passes vertically through the torus occipitalis, vertebral spinous process, gluteal cleft, the medial center of both knee joints, and the center of the intercarpal [8]. Continuing to have a posture that deviates from the ideal posture shortens and stiffens or stretches the muscles in the anterior and posterior parts of the body; this causes a condition called crossed syndrome, which causes muscle weakness [9]. Crossed syndrome also occurs in young adults and middle-aged people, such as students and office workers. As a result, crossed syndrome causes poor posture in young adults. To improve poor posture that leads to crossed syndrome, it is important to stretch the shortened muscles and strengthen the weakened muscles.

For a body that has suffered muscle weakness, squat exercises focus on the development of both lower and upper body muscles [10]; these exercises can be performed relatively easily in any setting. Therefore, squat exercises are commonly used in many sports to enhance athletic performance, as well as in post-operative rehabilitation programs [11]. Despite the advantages of squat exercises, an unstable posture during exercise could damage the lower back or place undue pressure on the knees [12]. To eliminate these potential risks, Cho et al. devised a modified wall-side squat performed against a wall [13]. These squats emphasize lumbar stability over lower body strength. Wall-side squats with abdominal retraction technology are said to reduce the occurrence of excessive lumbar lordosis and pelvic anteversion [14], and Lee has reported that stimulating somatosensory receptors is effective for improving posture [15].

To improve posture, it is also important that the body learns the ideal posture. A sitting posture with the pelvis tilted backward increases chest flexion and forward displacement of the head [16]. Noro et al. showed that the pelvic tilt angle displayed by priests while using a zafu (a round cushion used in Zen meditation) to meditate was close to 0° [17]. In our laboratory, we have devised a method to correct the sitting posture alignment using a cylindrical towel (rolled towel) as a substitute for the zafu [18]. Thus, it is necessary to learn the ideal standing and sitting postures to improve posture.

There have been reports on posture changes due to improved physical functions and the development of tools based on various ergonomic approaches. However, it appears that no reports have verified the effects of posture improvement by combining physical function improvement and posture learning exercises. Furthermore, although studies have verified improvements in posture immediately after an intervention [15], few studies have verified the sustained effects after an intervention. Therefore, the purpose of this study was to verify the effects of physical function improvement training and ideal posture learning exercises on the change in posture and determine whether these effects are sustained.

## 2. Materials and Methods

### 2.1. Subjects

Forty healthy female university students (aged 20–23 years) were enrolled. Their health history was unremarkable and their medications were not monitored. We determined the sample size based on a previous report [15]. The exclusion criteria were spinal or lower limb disease or pain, inability to exercise for any reason, and voluntarily engaging in sports or exercise more than two days a week. This study was approved by the ethics committees of our institutions (approval number: 2018-024). Written informed consent was obtained from subjects before participation in this study.

### 2.2. Measurements

#### 2.2.1. Basic Information

The age, height, weight, body mass index (BMI), and medical history were checked before the intervention.

#### 2.2.2. Posture Evaluation

In this study, a standing posture with the heels, buttocks, both scapulae, and torus occipitalis against the wall was defined as the ideal posture [8]. Posture angles were measured using a tilt angle measuring device (HORIZON, Yuki Trading; Tokyo, Japan) [19]. The trunk anteroposterior inclination (TAPI) and trunk left/right flexion (TLRF) were measured using the upper and lower edges of the sternum as landmarks [20]. The right anterior superior iliac spine (ASIS) and posterior superior iliac spines (PSIS) were used as landmarks to measure the pelvic anteroposterior inclination (PAPI), and the left and right ASIS were used as landmarks for measuring the pelvic left/right inclination (PLRI) and left/right rotation (PLRR) angles. The kyphosis angle was used as an index for measuring kyphosis [21]. The vertebral kyphosis angle (VKA) was calculated from the angles of inclination of the line connecting the 7th cervical spinal process and the maximum posterior region, and the line connecting the maximum posterior region and the midpoint of the superior iliac spines (Figure 1). To measure the pre-intervention posture, the subjects were instructed to stand in their usual posture.

### 2.3. Grouping

The 40 subjects were randomly and evenly assigned to one of the following 4 groups: Control group: Subjects did not perform any training or posture learning. Training group: Subjects performed a squat facing from a wall (“wall-side squatting”) and a stretching exercise with their hands placed on the wall (“wall-side stretching”). Posture learning group: Subjects performed an upright standing posture with their back against a wall (“standing upright with their back against the wall”) and a sitting posture in which they placed the back half of their buttocks on a rolled towel to support a neutral pelvis (“rolled towel under the sacrum”) [18]. Combination group: Subjects implemented both interventions of the training and posture learning groups.

### 2.4. Details of the 4-Week Program

#### 2.4.1. Wall-Side Squatting

Our devised wall-side squatting had two advantages over normal squats. One is that the knee protrudes forward more than the toe when squatting during normal squats but standing against the wall prevents the knee from protruding forward, reducing the burden on the knee. The second advantage is the ease of stimulating the posture-holding muscles as antigravity muscles, such as the spine erector, gluteal, and quadriceps muscles.

The subjects stood facing a wall with their feet spread apart at shoulder width and placed their toes against the wall with both feet abducted at 30°. Their hands were placed on the back of their head. The subjects then flexed their knee joints from 70° to 90° while inhaling for 3 s and extended their knee joints while exhaling for 3 s. During this time, they were reminded to keep their head facing forward and to keep their knees from bowing inward or outward. In addition, they were reminded to keep their knees from touching the wall (Figure 2a). A single flexion of the knee joints followed by an extension of the knee joints was counted as one motion, and ten motions were counted as one set.

#### 2.4.2. Wall-Side Stretching

Subjects put their hands on the left and right walls of a corner while standing and then put their right leg forward. While keeping their right knee flexed, they tilted forward as much as possible for 10 s while keeping their left heel on the floor. Subjects took care to stretch the right rectus femoris and left gastrocnemius. Next, both elbow joints were flexed so that the scapulae on both sides were brought to the midline, and then they were extended. This motion was performed five times. Subjects took care to stretch the pectoralis major, pectoralis minor, and serratus anterior. Finally, the right knee was extended, the head and neck were flexed, the trunk was flexed forward, and the pelvis was tilted backward for 10 s. After the head and neck were extended, the trunk was extended as well, and the pelvis was tilted forward for 10 s. In the first half, subjects took care to stretch their hamstrings, trapezius, thoracolumbar extensors, and deep neck muscles. In the second half, subjects took care to stretch the sternocleidomastoid, rectus abdominus, and iliopsoas. The stretches were then performed with the left and right legs switched (Figure 2b). Completion of the stretches with the left leg followed by the right leg was counted as one set.

#### 2.4.3. Standing Upright with Their Back against the Wall

Subjects stood with their back against a wall, kept their heels together, and pressed their heels to the wall. They then pulled their chin in and stretched their chest without extending their lumbar region. In this state, they put both their shoulders against the wall and took an upright position. Ten seconds in this position was counted as one set (Figure 2c).

#### 2.4.4. Rolled Towel under the Sacrum

A bath towel was rolled into a cylinder and placed against the back of a chair. Subjects sat down on the chair with the back half of their buttocks placed on the rolled towel. This towel prevents the pelvis from tilting backward, which is likely to occur in the sitting posture, and to grasp the sitting posture in the middle position of the pelvis. They assumed the sitting posture while making sure that their pelvis was neutral [18] (Figure 2d).

### 2.5. Measurement Protocol

The program was explained to the subjects on the first day. After collecting basic information from the subjects, their pre-intervention and ideal postures were measured. Afterward, the exercises performed against a wall (i.e., wall-side squatting, stretching, and standing upright with the back against the wall) were performed for 3 sets each and the rolled towel intervention was performed for 1 h. The subjects’ postures were measured immediately after completing their respective tasks. Subsequently, each subject performed each intervention exercise; the exercises against a wall were performed for 3 days a week at 3 sets a day for 4 weeks, and the rolled towel intervention was performed for at least 1 h every day for 4 weeks. The effect of the intervention was measured immediately after the first intervention on the first day, as well as on the last day of the 4-week intervention (day 27). The sustained effect of the intervention was measured 1 week after the last day of the intervention (day 34).

### 2.6. Definition of Improved Posture

In this study, improved posture was defined as the post-intervention posture approaching the measured values of the ideal posture. If the value obtained by subtracting the posture angle before intervention from the posture angle after intervention exceeded 0°, we considered that improved posture had been achieved (Figure 3).

### 2.7. Statistical Analysis

ANOVA was used to compare basic information between the groups. For the differences in improvement in the posture evaluation items between the groups, covariance analysis, in which the baseline measurement is used as the covariate, and the Bonferroni multiple comparison tests were used to compare groups. Data were analyzed using SPSS Statistics for Windows, ver. 25.0 (IBM, Armonk, NY, USA). The level of significance was set at *p* < 0.05.

## 3. Results

### 3.1. Basic Characteristics of Each Group

There were no significant differences between the basic characteristics of subjects in each group (Table 1). All subjects completed follow-up until the final measurement, and the dropout rate was 0%. There were no significant differences between groups in terms of the TLRF, PLRI, or PLRR angles before the intervention.

### 3.2. Comparison of Improvement in Posture Evaluation Items between Groups

Table 2 compares the improvement in posture evaluation items between groups. The TAPI significantly improved in the training (*p* = 0.009) and combination groups (*p* = 0.022). The PAPI significantly improved in the combination group (*p* < 0.001). On day 27, the combination group showed a significant improvement compared to the other intervention groups (*p* = 0.003, *p* = 0.022). The VKA was significantly improved among all intervention groups (*p* < 0.001, *p* < 0.001, *p* = 0.010), but none of the intervention groups showed sustained effects on this parameter.

### 3.3. Percentage of People with Improved Posture

Table 3 shows the percentages of subjects whose posture improved in terms of each posture evaluation and all three assessments. Immediately after the initial intervention, over 90% of the training group subjects improved in all posture evaluation items. On day 27, after the 4-week intervention, all subjects in the combination group improved in all posture evaluation items. Of the 30 subjects who underwent the intervention, 24 subjects (80%) improved in all items immediately after 4 weeks, and 8 subjects (26.6%) showed improvement in all items until day 34.

## 4. Discussion

The TAPI significantly improved in the training and combination groups. Therefore, it is suggested that wall-side squatting and stretching performed in the correct posture were effective in stretching and improving posture-holding muscle strength. Rancour et al. reported that if stretching is stopped completely, flexibility declines, but it can be maintained by continuing to stretch 2–3 days a week [22]. In this study, the intervention frequency of wall-side stretching was 3 days a week, and the improvement during the 4-week intervention period and the decrease in the improvement rate during the discontinuation period after the intervention support the findings by Rancour et al. After 4 weeks of continuous training, posture-holding muscle strength improved, and the improvement in muscle flexibility was sustained. However, improving postural muscles requires regular repetition training as well.

The combination group showed significantly greater improvement in the PAPI compared to the training and posture learning groups. It was suggested that the ideal angle of the pelvis could not be discerned through a squat or stretch intervention alone, nor could the ideal angle of the pelvis in a standing position be learned through the rolled towel intervention in a sitting position alone. Johnson et al. reported that an anterior pelvic tilt shortens the lumbar extensor muscles and extends the rectus abdominal and hip extensor muscles, while a posterior pelvic tilt causes the hip extensor muscles to shorten and the hip flexor muscles to extend [23]. Janda reported that lower crossed syndrome is characterized by the facilitation of the thoracolumbar extensors, rectus femoris, and iliopsoas, as well as inhibition of the abdominals (particularly transversus abdominus) and the gluteal muscles [24]. Therefore, combining the rolled towel intervention (to learn the ideal posture of the pelvis) and focusing training intervention on muscle strength (such as iliopsoas, abdominals, and gluteal muscles) was important for the persistence of learning proper posture.

The VKA was significantly improved in all intervention groups on day 27, but by day 34, the effects were not sustained in all groups. Claeys et al. reported that the VKA in the standing position is significantly correlated with the thoracic tilt angle, but not with the PAPI [25]. On the other hand, other studies reported moderate to high correlations between the pelvic and lumbar angles in the usual standing position [26,27,28]. Therefore, a consistent view of the relationship between the pelvis and spine was not obtained, and the results of this study did not determine which intervention program was effective.

In this study, the 4-week intervention showed improvement in posture, but no sustained effect at day 34. The change in posture-holding muscles could not be maintained using only the wall exercises in this study, and we suggest that it is important to continue the wall exercises regularly, even if only intermittently.

In addition, from immediately after a single intervention to one week later, only 27% of the subjects showed sustained effects in all three measurements. It is suggested that daily training is important for improving posture and that strengthening the posture-holding muscles requires regular repetition of exercises as well as the planning of programs tailored to each subject.

The present study has several limitations. First, we conducted a study on young females in reference to previous studies [15], but it is not clear whether the results of this study can be generalized to men or the elderly. Second, the effectiveness of providing appropriate programs to individual subjects has not been verified. The pre-intervention posture varied among groups, and it was not possible to verify which intervention was most effective for each subject’s poor posture. Third, the measurement methods in this study may not completely reflect all body posture changes due to kinematic chains. According to posture classifications such as those reported by Staffel and Kendall, poor postures vary, and the causes of each poor posture differ greatly [29,30]. For example, the swayback posture proposed by Kendall is a balance posture in which excessive back tilting of the pelvis is compensated for by kyphosis of the thoracolumbar spine [30]. In this posture, if the retrograde pelvis is improved by our intervention, the change in the pelvic longitudinal inclination can be verified. However, with respect to improved VKA, it is difficult to determine whether the posture has been improved by either (a) the effect of an improved PAPI angle or (b) by the wall-side upright intervention.

In the future, it will be necessary to investigate improvements in posture that are achieved when an appropriate and personalized intervention program is provided to each subject based on their pre-intervention posture. Furthermore, a larger sample size is necessary, as is required when using an evaluation method that captures whole-body posture changes such as X-ray imaging.

## 5. Conclusions

The combination of devised muscle training and posture learning effectively improved posture. However, it was found a week after the 4-week intervention, the subjects’ postures returned to their original postures (those before the intervention). For subjects to maintain an ideal posture, the proposed intervention needs to be continued and verified.

## Figures and Tables

**Figure 1 healthcare-11-01287-f001:**
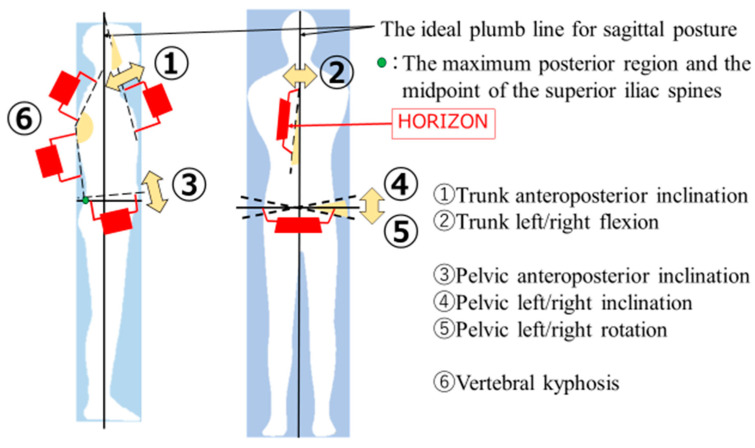
Posture evaluation.

**Figure 2 healthcare-11-01287-f002:**
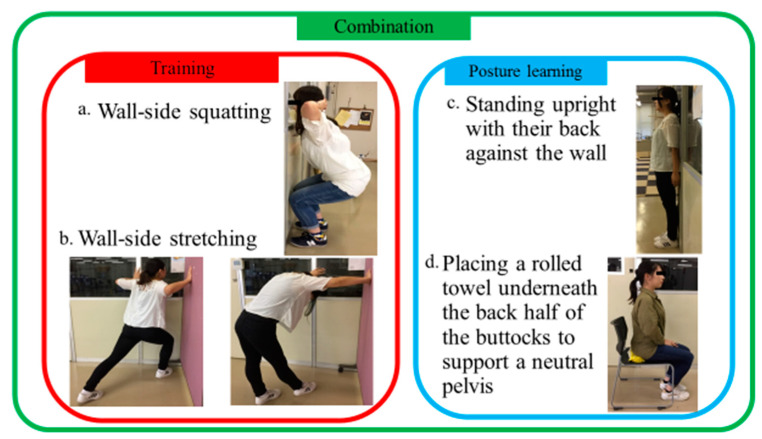
Grouping and details of the 4-week programs.

**Figure 3 healthcare-11-01287-f003:**
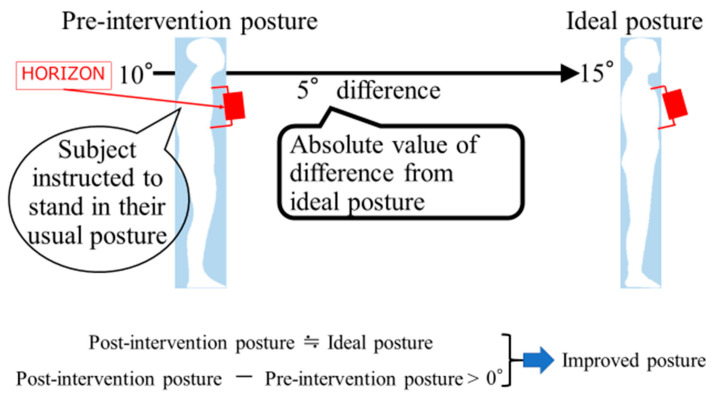
Definition of posture improvement.

**Table 1 healthcare-11-01287-t001:** Basic characteristics of each intervention group.

		Control (*n* = 10)	Training (*n* = 10)	Posture (*n* = 10)	Combination (*n* = 10)	F-Value	*p*-Value
Age (years)		20.3 ± 0.6	21.1 ± 0.9	21.1 ± 0.9	21.2 ± 0.7	2.30	0.09
Height (cm)		158.2 ± 5.5	159.8 ± 6.2	159.6 ± 4.9	159.8 ± 3.2	0.20	0.89
Weight (kg)		52.5 ± 7.1	54.8 ± 6.6	53.4 ± 6.2	55.4 ± 6.4	0.37	0.77
BMI (kg/m^2^)		21.0 ± 1.7	21.4 ± 1.7	20.9 ± 1.7	21.7 ± 2.0	0.41	0.74
TAPI (°)	Ideal	19.2 ± 3.5	17.9 ± 2.7	17.0 ± 3.8	16.8 ± 3.3	1.08	0.37
	Pre	15.7 ± 3.3	12.5 ± 3.6	15.9 ± 2.5	12.5 ± 3.3	3.59	0.02
PAPI (°)	Ideal	10.9 ± 1.8	15.2 ± 2.3	12.0 ± 2.1	16.4 ± 1.6	17.50	0.00
	Pre	6.6 ± 0.9	9.5 ± 3.4	6.8 ± 1.7	9.2 ± 2.8	4.08	0.01
VKA (°)	Ideal	170.8 ± 3.4	170.4 ± 3.7	169.5 ± 5.6	170.8 ± 3.2	0.21	0.89
	Pre	166.4 ± 2.9	163.3 ± 4.3	164.1 ± 7.0	165.9 ± 4.7	0.86	0.47

Mean ± SD; BMI: body mass index; TAPI: trunk anteroposterior inclination; PAPI: pelvic anteroposterior inclination; VKA: vertebral kyphosis angle.

**Table 2 healthcare-11-01287-t002:** Comparison of improvement in posture evaluation items between groups.

		Pairs Compared		No Adjustment					Adjustment ^a^				
				Average ^b^	95% CI			*p*-Value	Average ^b^	95% CI			*p*-Value
TAPI	Post day 1	Control	Training	−3.40	−6.18	to	−0.62	0.009	−2.29	−4.82	to	0.25	0.097
			Posture	−0.17	−2.94	to	2.61	1.000	−1.18	−3.88	to	1.51	1.000
			Combination	−2.53	−5.31	to	0.24	0.091	−2.18	−4.75	to	0.39	0.139
		Combination	Training	−0.87	−3.64	to	1.91	1.000	−0.11	−2.67	to	2.45	1.000
			Posture	2.37	−0.41	to	5.14	0.136	1.00	−1.73	to	3.72	1.000
		Posture	Training	−3.23	−6.01	to	−0.46	0.015	−1.11	−3.80	to	1.58	1.000
	Day 27	Control	Training	−3.80	−6.46	to	−1.14	0.002	−2.55	−4.62	to	−0.49	0.009
			Posture	0.27	−2.39	to	2.92	1.000	−0.64	−2.84	to	1.56	1.000
			Combination	−2.90	−5.56	to	−0.24	0.026	−2.34	−4.43	to	−0.25	0.022
		Combination	Training	−0.90	−3.56	to	1.76	1.000	−0.21	−2.30	to	1.88	1.000
			Posture	3.17	0.51	to	5.82	0.012	1.70	−0.52	to	3.92	0.232
		Posture	Training	−4.07	−6.72	to	−1.41	0.001	−1.91	−4.11	to	0.28	0.118
	Day 34	Control	Training	−1.70	−4.30	to	0.90	0.458	−0.49	−2.38	to	1.40	1.000
			Posture	1.10	−1.50	to	3.70	1.000	0.52	−1.49	to	2.53	1.000
			Combination	−1.00	−3.60	to	1.60	1.000	0.03	−1.88	to	1.95	1.000
		Combination	Training	−0.70	−3.30	to	1.90	1.000	−0.52	−2.43	to	1.39	1.000
			Posture	2.10	−0.50	to	4.70	0.182	0.49	−1.54	to	2.52	1.000
		Posture	Training	−2.80	−5.40	to	−0.20	0.029	−1.01	−3.02	to	1.00	1.000
PAPI	Post day 1	Control	Training	−2.43	−4.91	to	0.04	0.056	−2.49	−6.02	to	1.05	0.338
			Posture	−3.20	−5.68	to	−0.72	0.006	−2.47	−6.11	to	1.17	0.389
			Combination	−4.77	−7.24	to	−2.29	<0.001	−5.35	−8.87	to	−1.83	<0.001
		Combination	Training	2.33	−0.14	to	4.81	0.075	2.86	0.47	to	5.26	0.012
			Posture	1.57	−0.91	to	4.04	0.514	2.87	0.32	to	5.42	0.020
		Posture	Training	0.77	−1.71	to	3.24	1.000	−0.01	−2.58	to	2.56	1.000
	Day 27	Control	Training	−2.43	−5.05	to	0.18	0.081	−2.93	−6.44	to	0.58	0.151
			Posture	−4.03	−6.65	to	−1.42	<0.001	−3.36	−6.98	to	0.25	0.081
			Combination	−5.30	−7.91	to	−2.69	<0.001	−6.19	−9.68	to	−2.69	<0.001
		Combination	Training	2.87	0.25	to	5.48	0.025	3.26	0.88	to	5.64	0.003
			Posture	1.27	−1.35	to	3.88	1.000	2.82	0.29	to	5.36	0.022
		Posture	Training	1.60	−1.01	to	4.21	0.577	0.43	−2.12	to	2.99	1.000
	Day 34	Control	Training	−1.10	−2.72	to	0.52	0.393	−1.49	−3.20	to	0.22	0.121
			Posture	−0.30	−1.92	to	1.32	1.000	0.36	−1.40	to	2.13	1.000
			Combination	−1.47	−3.08	to	0.15	0.095	−1.76	−3.47	to	−0.05	0.040
		Combination	Training	0.37	−1.25	to	1.98	1.000	0.27	−0.89	to	1.43	1.000
			Posture	1.17	−0.45	to	2.78	0.309	2.12	0.89	to	3.36	<0.001
		Posture	Training	−0.80	−2.42	to	0.82	1.000	−1.85	−3.10	to	−0.61	0.001
VKA	Post day 1	Control	Training	−3.77	−7.19	to	−0.34	0.024	−2.23	−4.92	to	0.46	0.156
			Posture	−3.93	−7.36	to	−0.51	0.017	−3.12	−5.72	to	−0.51	0.012
			Combination	−2.10	−5.52	to	1.32	0.572	−1.77	−4.40	to	0.85	0.398
		Combination	Training	−1.67	−5.09	to	1.76	1.000	−0.46	−3.02	to	2.10	1.000
			Posture	−1.83	−5.26	to	1.59	0.862	−1.35	−3.82	to	1.13	0.815
		Posture	Training	0.17	−3.26	to	3.59	1.000	0.89	−1.66	to	3.43	1.000
	Day 27	Control	Training	−4.87	−8.03	to	−1.70	<0.001	−3.50	−5.66	to	−1.33	<0.001
			Posture	−4.40	−7.57	to	−1.23	0.003	−3.64	−5.73	to	−1.54	<0.001
			Combination	−2.73	−5.90	to	0.43	0.127	−2.57	−4.68	to	−0.46	0.010
		Combination	Training	−2.13	−5.30	to	1.03	0.409	−0.92	−2.98	to	1.14	1.000
			Posture	−1.67	−4.83	to	1.50	0.903	−1.06	−3.05	to	0.93	0.856
		Posture	Training	−0.47	−3.63	to	2.70	1.000	0.14	−1.91	to	2.18	1.000
	Day 34	Control	Training	−1.97	−5.16	to	1.23	0.567	−0.67	−3.28	to	1.95	1.000
			Posture	−1.90	−5.10	to	1.30	0.635	−1.62	−4.15	to	0.91	0.486
			Combination	−1.37	−4.56	to	1.83	1.000	−1.49	−4.04	to	1.06	0.665
		Combination	Training	−0.60	−3.80	to	2.60	1.000	0.82	−1.67	to	3.31	1.000
			Posture	−0.53	−3.73	to	2.66	1.000	−0.14	−2.54	to	2.27	1.000
		Posture	Training	−0.07	−3.26	to	3.13	1.000	0.96	−1.51	to	3.43	1.000

CI: confidence interval; Post day 1: after the first intervention; Day 27: the last day of the 4-week intervention; Day 34: one week after the last day of the intervention; TAPI: trunk anteroposterior inclination; PAPI: pelvic anteroposterior inclination; VKA: vertebral kyphosis angle; ^a^ adjusted for each pre-value (TAPI, PAPI, VKA); ^b^ mean difference between paired groups.

**Table 3 healthcare-11-01287-t003:** Percentage of subjects with improved posture.

	Control (*n* = 10)	Training (*n* = 10)	Posture (*n* = 10)	Combination (*n* = 10)
Post day 1	20/40/50/10	90/90/90/70	70/90/90/60	90/100/80/70
Day 27	40/10/40/0	100/90/100/90	60/100/90/50	100/100/100/100
Day 34	30/10/50/0	70/70/70/40	50/30/80/20	50/60/60/20

TAPI (%)/PAPI (%)/VKA (%)/all three items (%). Post day 1: after the first intervention; Day 27: the last day of the 4-week intervention; Day 34: one week after the last day of the intervention; TAPI: trunk anteroposterior inclination; PAPI: pelvic anteroposterior inclination; VKA: vertebral kyphosis angle.

## Data Availability

Research subjects were asked to sign a document regarding consent and withdrawal of data use.

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
