# Peer review of "Effect of Devised Simultaneous Physical Function Improvement Training and Posture Learning Exercises on Posture"

_healthcare, 2023, doi:10.3390/healthcare11091287_

Round 1

Reviewer 1 Report (New Reviewer)

Dear Author

This was an interesting study that attempts to improve posture.

Please reiterate the following matters.

[Background]

-        The subject of this study was young female. The first paragraph, please describe more clinical problems caused by poor posture in young people rather than falls in the elderly.

[Materials and methods]

-        Have you demonstrated the reproducibility and validity of your postural measurement methods?

-        How were the effects of shoes and clothing considered in the postural measurements?

[Discussion]

-        Please describe the significance of your research by improving the posture of young female. How about including a citation for shoulder pain or back pain in women?

Author Response

April 27, 2023

Dear Reviewer:

Thank you for your valuable advice.

I will reply below.

[Background]

- The subject of this study was young female. The first paragraph, please describe more clinical problems caused by poor posture in young people rather than falls in the elderly.

➡The first paragraph has been corrected. Thank you for pointing this out.

[Materials and methods]

- Have you demonstrated the reproducibility and validity of your postural measurement methods?

➡Reproducibility and validity have not been demonstrated.This study was conducted with reference to the measurement methods used in previous studies.

- How were the effects of shoes and clothing considered in the postural measurements?

➡We also did not take into account the effects of shoes and clothing.

However, we believe that the influence of shoes can be dismissed because we measured the posture of the subjects barefoot by the wall.

The clothing design was not standardized, and most of the female students wore sweatshirts or comfortable everyday clothes because they were students.

[Discussion]

- Please describe the significance of your research by improving the posture of young female. How about including a citation for shoulder pain or back pain in women?

➡Thank you for pointing out the significance of this study for young women, the subject of this study, in the Introduction.

Thank you for your consideration. I look forward to hearing from you.

Sincerely,

First author: Sho Kudo

[Affiliation] Sensory and Motor Control, Graduate School of Medical Sciences, Kitasato University

[Postal address] 1-15-1 Kitasato, Minami, Sagamihara, Kanagawa 252-0373, Japan

[Phone/Fax number] +81-42-778-8111

[Email address] [email protected]

Corresponding author: Naonobu Takahira

[Affiliation] Physical Therapy Course, Department of Rehabilitation, School of Allied Health Sciences, Kitasato University

[Postal address] 1-15-1, Kitasato Minami, Sagamihara, Kanagawa 252-0373, Japan

[Phone/Fax number] +81-42-778-8111

[Email address] [email protected]

Reviewer 2 Report (New Reviewer)

This manuscript reports on a randomized controlled trial involving physical function improvement training and posture learning educational interventions for correcting poor posture. Postural measurements were obtained pre-trial, post-trial, and at 4-week follow-up and compared within and between 4 groups: control group, physical function improvement training, posture learning, and combined physical function improvement training with posture learning. Combined physical function improvement training with posture learning showed an effective improvement in all postural alignments. Four-week follow-up assessment after interventions ceased showed postures returned to pre-trial baseline.

The authors did a good job of organizing this study and provided a well-written and organized manuscript to present their information, data, and conclusions.

Page 1, Lines 11-12

"while poor posture in young adults and middle-aged people is due to crossed syndrome."

This is not an accurate statement. Studies showing a link between poor spinal posture and crossed syndromes have not identified the crossed syndrome as a cause. Postural changes may very well be the causative agent in the change in muscular biomechanics. Additionally, the first part of the sentence is stating what ill-health effects can come from poor posture in elderly populations. It would make more sense to keep the same focus in the latter part of the sentence. An example would be:

Poor posture in the elderly is associated with falls, while poor posture in young adults and middle-aged people is associated with neck and back pain which are among the leading causes of disability worldwide.

You address this more in your introduction but go back and forth between statements that indicate that posture causes crossed syndrome (such as this statement in lines 41-44: "Continuing to have a posture that deviates from the ideal posture shortens and stiffens or stretches the muscles in the anterior and posterior parts of the body; this causes a condition called crossed syndrome, which causes muscle weakness" and this statement in line 46: "To improve poor posture which leads to crossed syndrome...") and statements that indicate that crossed syndrome cause poor posture (such as this statement in lines 45-46: "...crossed syndrome causes poor posture in young adults."

As stated previously, studies showing a link between poor spinal posture and crossed syndromes have not identified the crossed syndrome as a cause. Postural changes may very well be the causative agent in the change in muscular biomechanics. It makes more sense to describe the poor posture and crossed syndromes as being correlated/linked to each other and that helping one variable can help the other. But please find a reference for any statements that indicate this.

Lines 66-67

Please provide references for the following statement:

"There have been reports on posture changes due to improved physical functions and 66 the development of tools based on various ergonomic approaches."

Lines 70-71

The following statement needs to be reconsidered:

"...few studies have verified the sustained effects after an intervention."

Below are references that refute this statement:

https://pubmed.ncbi.nlm.nih.gov/23640324/ (6-month follow-up)

https://pubmed.ncbi.nlm.nih.gov/22684211/ (6-month follow-up)

https://pubmed.ncbi.nlm.nih.gov/22632584/ (3-month follow-up)

https://www.sciencedirect.com/science/article/pii/S0031940611000691 (page 829, bottom right, 12-week follow-up)

https://www.mdpi.com/2077-0383/11/19/5768 (6-month follow-up)

https://pubmed.ncbi.nlm.nih.gov/27576192/ (1-year follow-up)

https://pubmed.ncbi.nlm.nih.gov/27575013/ (1-year follow-up)

https://pubmed.ncbi.nlm.nih.gov/28372313/ (10-week follow-up)

https://pubmed.ncbi.nlm.nih.gov/30419868/ (1-year follow-up)

https://pubmed.ncbi.nlm.nih.gov/33543266/ (1-year follow-up)

https://www.mdpi.com/2077-0383/11/21/6515 (2-year follow-up)

https://www.mdpi.com/2077-0383/11/20/6028 (1-year follow-up)

https://www.mdpi.com/2077-0383/11/19/5768 (6-month follow-up)

https://www.mdpi.com/2077-0383/12/2/542 (3-month follow-up)

Line 91

"The trunk anteroposterior inclination and left/right flexion (TAPI, TLRF) were measured..."

This should be rewritten as:

The trunk anteroposterior inclination (TAPI) and trunk left/right flexion (TLRF) were measured...

Line 92-93

"The right anterior and posterior superior iliac spines..."

This should be rewritten as:

The right anterior superior iliac spine (ASIS) and posterior superior iliac spine (PSIS)...

Then use ASIS and PSIS (known abbreviations) for the remainder of the manuscript.

 Line 95-96

"...for measuring the pelvic left/right inclination and left/right rotation angles (PLRI, PLRR)."

This should be rewritten as:

...for measuring the pelvic left/right inclination (PLRI) and pelvic left/right rotation (PLRR) angles.

Lines 96-100

The explanation of how the kyphosis angle was measured is confusing, especially considering the left image in Figure 1. The upper line of the kyphosis angle measurement looks good, but the lower line looks like it is based on the vertical solid line, which needs to be labeled (I assume this is the ideal plumb line for sagittal posture). It does not seem to connect to the "midpoint of the superior iliac spines" nor could it do so in a sagittal view as the midpoint of the superior iliac spines is visible on the coronal view.

I have seen VKA measured using posture as the following:

Vertebral kyphosis angle (VKA) is calculated from the angles of inclination of the line connecting the 1st thoracic spinal process and the maximum posterior region, and the line connecting the maximum posterior region and the 12th thoracic spinal process.

The Ueda T, et al paper you referenced assessed the validity of the measurements you used for VKA against the flexicurve ruler, which is not the standard for thoracic kyphosis measurements. Instead, this measurement needs to be validated according to lateral thoracic radiography with the participant in a standing/upright, weight-bearing position.

The two images in Figure 1 should be A and B so as to better reference and understand.

Lines 118-119

Please provide a reference for the following statement:

"...standing against the wall prevents the knee from protruding forward, reducing the burden on the knee."

The quality of Figure 1 is low and that of Figures 2 and 3 are very low. Please provide a high-quality image for clarity.

Please provide full text for reference 11 to confirm the following statements from lines 53-58:

"To eliminate these potential risks, Cho et al. devised a modified wall-side squat performed against a wall [10]. These squats emphasize lumbar stability over lower body strength. Wall-side squats with abdominal retraction technology are said to reduce the occurrence of excessive lumbar lordosis and pelvic anteversion [11], and Lee has reported that stimulating somatosensory receptors is effective for improving posture [12]."

Please provide full text for reference 18 to confirm the following statements from lines 96-100:

"The kyphosis angle was used as an index for measuring kyphosis [18]. The vertebral kyphosis angle (VKA) was calculated from the angles of inclination of the line connecting the 7th cervical spinal process and the maximum posterior region, and the line connecting the maximum posterior region and the midpoint of the superior iliac spines (Figure 1)."

It looks like there is a 27th reference that is not numbered:

Kendall FP, McCreary EK, Provance PG, Rodgers MM, Romani WA, Muscles: Testing and Function with Posture and Pain, 5th Edition., Lippincott Williams & Wilkins, Philadelphia, 2005.

References 15, 20, 21, 26, and 27 do not have links.

In future studies, it would help to provide additional metrics including functional (ie range of motion or balance) and symptomatic/participant subjective (ie quality of life, pain, or disability questionnaires and outcomes) measurements to substantiate the health effects of improving posture.

Author Response

April 27, 2023

Dear Reviewer:

Thank you for your valuable advice.

I will reply below.

Page 1, Lines 11-12

"while poor posture in young adults and middle-aged people is due to crossed syndrome."

This is not an accurate statement. Studies showing a link between poor spinal posture and crossed syndromes have not identified the crossed syndrome as a cause. Postural changes may very well be the causative agent in the change in muscular biomechanics. Additionally, the first part of the sentence is stating what ill-health effects can come from poor posture in elderly populations. It would make more sense to keep the same focus in the latter part of the sentence. An example would be:

Poor posture in the elderly is associated with falls, while poor posture in young adults and middle-aged people is associated with neck and back pain which are among the leading causes of disability worldwide.

You address this more in your introduction but go back and forth between statements that indicate that posture causes crossed syndrome (such as this statement in lines 41-44: "Continuing to have a posture that deviates from the ideal posture shortens and stiffens or stretches the muscles in the anterior and posterior parts of the body; this causes a condition called crossed syndrome, which causes muscle weakness" and this statement in line 46: "To improve poor posture which leads to crossed syndrome...") and statements that indicate that crossed syndrome cause poor posture (such as this statement in lines 45-46: "...crossed syndrome causes poor posture in young adults."

As stated previously, studies showing a link between poor spinal posture and crossed syndromes have not identified the crossed syndrome as a cause. Postural changes may very well be the causative agent in the change in muscular biomechanics. It makes more sense to describe the poor posture and crossed syndromes as being correlated/linked to each other and that helping one variable can help the other. But please find a reference for any statements that indicate this.

➡Editer➀ also provided feedback, and we have revised the text based on both of their opinions. Thank you for pointing this out.

Lines 66-67

Please provide references for the following statement:

"There have been reports on posture changes due to improved physical functions and the development of tools based on various ergonomic approaches."

➡The reference is in Japanese, and I am currently searching Pubmed for references cited in English papers. We will post references that are appropriate for the description.

Lines 70-71

The following statement needs to be reconsidered:

"...few studies have verified the sustained effects after an intervention."

Below are references that refute this statement:

https://pubmed.ncbi.nlm.nih.gov/23640324/ (6-month follow-up)

https://pubmed.ncbi.nlm.nih.gov/22684211/ (6-month follow-up)

https://pubmed.ncbi.nlm.nih.gov/22632584/ (3-month follow-up)

https://www.sciencedirect.com/science/article/pii/S0031940611000691 (page 829, bottom right, 12-week follow-up)

https://www.mdpi.com/2077-0383/11/19/5768 (6-month follow-up)

https://pubmed.ncbi.nlm.nih.gov/27576192/ (1-year follow-up)

https://pubmed.ncbi.nlm.nih.gov/27575013/ (1-year follow-up)

https://pubmed.ncbi.nlm.nih.gov/28372313/ (10-week follow-up)

https://pubmed.ncbi.nlm.nih.gov/30419868/ (1-year follow-up)

https://pubmed.ncbi.nlm.nih.gov/33543266/ (1-year follow-up)

https://www.mdpi.com/2077-0383/11/21/6515 (2-year follow-up)

https://www.mdpi.com/2077-0383/11/20/6028 (1-year follow-up)

https://www.mdpi.com/2077-0383/11/19/5768 (6-month follow-up)

https://www.mdpi.com/2077-0383/12/2/542 (3-month follow-up)

➡Thank you for suggesting numerous references. There were a few references that I was not able to access in full, but I am currently reviewing the ones you suggested.

 I also believe that the primary significance of this study is that it examines the effectiveness of a combined stretching and training intervention to improve poor posture. I hope you will find it useful in comparison with the previous studies you mentioned.

Thank you for your suggestion.

Line 91

"The trunk anteroposterior inclination and left/right flexion (TAPI, TLRF) were measured..."

This should be rewritten as:

The trunk anteroposterior inclination (TAPI) and trunk left/right flexion (TLRF) were measured...

➡Thank you for pointing this out. We have corrected it.

Line 92-93

"The right anterior and posterior superior iliac spines..."

This should be rewritten as:

The right anterior superior iliac spine (ASIS) and posterior superior iliac spine (PSIS)...

Then use ASIS and PSIS (known abbreviations) for the remainder of the manuscript.

➡Thank you for pointing this out. We have corrected it.

 Line 95-96

"...for measuring the pelvic left/right inclination and left/right rotation angles (PLRI, PLRR)."

This should be rewritten as:

...for measuring the pelvic left/right inclination (PLRI) and pelvic left/right rotation (PLRR) angles.

➡Thank you for pointing this out. We have corrected it.

Lines 96-100

The explanation of how the kyphosis angle was measured is confusing, especially considering the left image in Figure 1. The upper line of the kyphosis angle measurement looks good, but the lower line looks like it is based on the vertical solid line, which needs to be labeled (I assume this is the ideal plumb line for sagittal posture). It does not seem to connect to the "midpoint of the superior iliac spines" nor could it do so in a sagittal view as the midpoint of the superior iliac spines is visible on the coronal view.

I have seen VKA measured using posture as the following:

Vertebral kyphosis angle (VKA) is calculated from the angles of inclination of the line connecting the 1st thoracic spinal process and the maximum posterior region, and the line connecting the maximum posterior region and the 12th thoracic spinal process.

The Ueda T, et al paper you referenced assessed the validity of the measurements you used for VKA against the flexicurve ruler, which is not the standard for thoracic kyphosis measurements. Instead, this measurement needs to be validated according to lateral thoracic radiography with the participant in a standing/upright, weight-bearing position.

The two images in Figure 1 should be A and B so as to better reference and understand.

➡Thank you for pointing this out. We have corrected Figure 1. If this figure is still difficult to interpret, I will create a new figure in the form of a separate figure for VKA only.

Lines 118-119

Please provide a reference for the following statement:

"...standing against the wall prevents the knee from protruding forward, reducing the burden on the knee."

➡With regard to this statement, the mechanical relationship between the load line and the center of gravity is described in consideration of the fact that the mechanical rotational moment to the knee joint increases with anterior protrusion of the knee joint. We made this statement based on consideration from a mechanical point of view rather than prior research.

The quality of Figure 1 is low and that of Figures 2 and 3 are very low. Please provide a high-quality image for clarity.

➡We have attached a new diagram. Please check the image quality.

Please provide full text for reference 11 to confirm the following statements from lines 53-58:

"To eliminate these potential risks, Cho et al. devised a modified wall-side squat performed against a wall [10]. These squats emphasize lumbar stability over lower body strength. Wall-side squats with abdominal retraction technology are said to reduce the occurrence of excessive lumbar lordosis and pelvic anteversion [11], and Lee has reported that stimulating somatosensory receptors is effective for improving posture [12]."

➡I have attached a PDF file. Please check it.

Please provide full text for reference 18 to confirm the following statements from lines 96-100:

"The kyphosis angle was used as an index for measuring kyphosis [18]. The vertebral kyphosis angle (VKA) was calculated from the angles of inclination of the line connecting the 7th cervical spinal process and the maximum posterior region, and the line connecting the maximum posterior region and the midpoint of the superior iliac spines (Figure 1)."

➡I have attached a PDF file. Please check it.

It looks like there is a 27th reference that is not numbered:

Kendall FP, McCreary EK, Provance PG, Rodgers MM, Romani WA, Muscles: Testing and Function with Posture and Pain, 5th Edition., Lippincott Williams & Wilkins, Philadelphia, 2005.

➡These documents are books and are not numbered.

References 15, 20, 21, 26, and 27 do not have links.

➡These documents are books and there are no links.

In future studies, it would help to provide additional metrics including functional (ie range of motion or balance) and symptomatic/participant subjective (ie quality of life, pain, or disability questionnaires and outcomes) measurements to substantiate the health effects of improving posture.

➡Thank you for pointing this out. We will take this feedback into consideration for future research.

Thank you for your consideration. I look forward to hearing from you.

Sincerely,

First author: Sho Kudo

[Affiliation] Sensory and Motor Control, Graduate School of Medical Sciences, Kitasato University

[Postal address] 1-15-1 Kitasato, Minami, Sagamihara, Kanagawa 252-0373, Japan

[Phone/Fax number] +81-42-778-8111

[Email address] [email protected]

Corresponding author: Naonobu Takahira

[Affiliation] Physical Therapy Course, Department of Rehabilitation, School of Allied Health Sciences, Kitasato University

[Postal address] 1-15-1, Kitasato Minami, Sagamihara, Kanagawa 252-0373, Japan

[Phone/Fax number] +81-42-778-8111

[Email address] [email protected]

This manuscript is a resubmission of an earlier submission. The following is a list of the peer review reports and author responses from that submission.

Round 1

Reviewer 1 Report

The topic is interesting and also applicable in clinical settings. However, some issues have to be revised.

Title – the word “our” in the title is unnecessary. I suggest that the authors remove this word from the title (The same in the conclusions).

Introduction - it is clearly described

Methods - The inclusion criteria need to be better explained, for example, did these participants have any previous heart, respiratory, neurological disease? did they use medication? why only female?

Results - The statistical analysis conducted is pervasive in their description.

Discussion - The discussions are very well-detailed towards a clinical description. 

Information is missing regarding the following topics, which should be at the end of the text: Funding, Institutional Review Board Statement, Informed Consent Statement, Data Availability Statement and Conflicts of Interest

Author Response

Thank you for your valuable advice.

I will reply below.

  • Title – the word “our” in the title is unnecessary. I suggest that the authors remove this word from the title (The same in the conclusions).

➡I fixed the wording. Thank you for your advice.

  • Methods - The inclusion criteria need to be better explained, for example, did these participants have any previous heart, respiratory, neurological disease? did they use medication? why only female?

➡Only healthy female university students were included in the study because they were based on previous research. No medications or medical history.

  • Information is missing regarding the following topics, which should be at the end of the text: Funding, Institutional Review Board Statement, Informed Consent Statement, Data Availability Statement and Conflicts of Interest

➡Thank you for your advice. We will add text.

Reviewer 2 Report

Review healthcare-2222469

First, I would like to thank you for the opportunity to review this manuscript.

The authors present a paper about effects of physical function improvement training and posture learning exercises on posture. They indicate to be the first who investigated sustaining effects of both interventions. Results suggest, that the interventions were effective particularly in combination, but have to be continued.

The topic is important, and the manuscript is mainly well written. However, I have some major remarks the authors should deal with:

General comments:

-          I could not find any references younger than 2016, most cited references were quite old. Ich suggest including some up-to-date literature.

-          I miss psychological factors. Ikemoto et al. (2016), for instance, found depression to be relevant regarding the topic of locomotive syndrome.

-          Further I miss information about pain, which usually is associated with crossed syndrome and locomotive syndrome.

-          Please provide a better quality of figures.

-          It is not clear to me, whether participants were advised to do no exercises after the four weeks of intervention. I interpreted so, but it is not described. Please complete.

-          The structure of the discussion is missing a thread.

Specific comments:

Lines 41ff (just a comment to consider)

The discussion about shortening and lengthening of muscles is still actual. There are heterogenous results about the possibility of lengthening and shortening of muscles and its structures. However, I suggest including terms such as “tonic” and “phasic” muscles in combination with a short explanation.

Line 106

Maybe I do not understand correctly, but as far as I understand the squat exercise, participants performed facing the wall and not “away”. If so, please revise.

Lines 130-145

It was difficult for me to understand immediately the procedure of the stretching exercise. I suggest revising this paragraph.

Specifically, I am sceptical if the right rectus femoris can be stretched when the right leg is flexed as demonstrated in the figure 2b.

Maybe more pictures of the different stages of the exercise could be helpful.

Line 140

“In the first half” – of what? Of the exercise or of 10 seconds?

Line 160

What do you mean with “pre-intervention”?

And how can you measure “ideal" postures before intervention?

Line 196

How do you define “training group” here? In paragraph 2.3 the training group is defined as group exercising squats and the stretching. At this point I am not sure, whether training groups includes posture learning as well, because I can not find results for this latter group in the text.

Discussion

I suggest to revise the structure in favour of a better reading flow.

Author Response

Thank you for your valuable advice.

I will reply below.

General comments:

  • I could not find any references younger than 2016, most cited references were quite old. Ich suggest including some up-to-date literature.
  • I miss psychological factors. Ikemoto et al. (2016), for instance, found depression to be relevant regarding the topic of locomotive syndrome.
  • Further I miss information about pain, which usually is associated with crossed syndrome and locomotive syndrome.
  • Please provide a better quality of figures.

➡Thank you for your advice. We are currently reviewing the literature. Regarding the figure, if it is the specified size, it will be of this quality.

  • It is not clear to me, whether participants were advised to do no exercises after the four weeks of intervention. I interpreted so, but it is not described. Please complete.

➡I fixed the wording. Thank you for your advice.

Specific comments:

Lines 41ff (just a comment to consider)

The discussion about shortening and lengthening of muscles is still actual. There are heterogenous results about the possibility of lengthening and shortening of muscles and its structures. However, I suggest including terms such as “tonic” and “phasic” muscles in combination with a short explanation.

➡Thank you for your advice. I will consult with Professor Takahira.

Line 106

Maybe I do not understand correctly, but as far as I understand the squat exercise, participants performed facing the wall and not “away”. If so, please revise.

➡I fixed the wording. Thank you for your advice.

Lines 130-145

It was difficult for me to understand immediately the procedure of the stretching exercise. I suggest revising this paragraph.

Specifically, I am sceptical if the right rectus femoris can be stretched when the right leg is flexed as demonstrated in the figure 2b.

Maybe more pictures of the different stages of the exercise could be helpful.

➡We will reconsider. Thank you for your advice.

Line 140

“In the first half” – of what? Of the exercise or of 10 seconds?

➡”In the first half” = Finally, the right knee was extended, the head and neck were flexed, the trunk was flexed forward, and the pelvis was tilted backward for 10 seconds.

Line 160

What do you mean with “pre-intervention”?

➡“pre-intervention”=”before intervention”

And how can you measure “ideal" postures before intervention?

➡In this study, a standing posture with the heels, buttocks, both scapulae, and the to-rus occipitalis against the wall was defined as the ideal posture [5]. Posture angles were measured using a tilt angle measuring device (HORIZON, Yuki Trading; Tokyo, Japan) [16].

Line 196

How do you define “training group” here? In paragraph 2.3 the training group is defined as group exercising squats and the stretching. At this point I am not sure, whether training groups includes posture learning as well, because I can not find results for this latter group in the text.

➡Excuse me. I cannot understand what you are pointing out.

 Training group=Subjects performed a squat facing from a wall (“wall-side squatting”) and a stretching exercise with their hands placed on the wall (“wall-side stretching”).

Reviewer 3 Report

The reported study is quite interesting and clearly described. Authors describe a study performed on a group of 40 young female students to verify the effects of physical function improvement training and ideal posture learning exercises on the change in posture and their sustained effects. The study was well conducted with the appropriate materials and methodologies. The findings are also clearly supported by the results. Since authors stated that one week after the four weeks program, some subjects tend to return to the pre-program posture, it would be interesting to study how long this posture regression takes to every subject in the study, or even if for some subjects the posture improvement is more long lasting.

Author Response

Thank you for your valuable advice. I am honored to receive such a positive response.